# Effect of Contact Plug Deposition Conditions on Junction Leakage and Contact Resistance in Multilevel CMOS Logic Interconnection Device

**DOI:** 10.3390/mi11020170

**Published:** 2020-02-06

**Authors:** Yinhua Cui, Jeong Yeul Jeong, Yuan Gao, Sung Gyu Pyo

**Affiliations:** 1School of Integrative Engineering, Chung-Ang University, Seoul 06974, Korea; yinhua0822@gmail.com (Y.C.); gaoyuan4025@gmail.com (Y.G.); 2Process Development Center, Magnachip Semiconductor, Seoul 15213, Korea; trinitysg@naver.com

**Keywords:** junction leakage, contact resistance, contact metallization

## Abstract

Here, we developed the optimal conditions in terms of physical and electrical characteristics of the barrier and tungsten (W) deposition process of a contact module, which is the segment connecting the device and the multi-layer metallization (MLM) metal line in the development of 100 nm-class logic devices. To confirm its applicability to the logic contact of barrier and W films, a contact hole was formed, first to check the bottom coverage and the filling status of each film, then to check the electrical resistance and leakage characteristics to analyze the optimal conditions. At an aspect ratio of 3.89:1, ionized metal plasma (IMP) Ti had a bottom coverage of 40.9% and chemical vapor deposition (CVD) titanium nitride (TiN) of 76.2%, confirming that it was possible to apply the process to 100 nm logic contacts. W filling was confirmed, and a salicide etching rate (using Radio Frequency (RF) etch) of 13–18 Å/s at a 3.53:1 aspect ratio was applied. The etching rate on the thermal oxide plate was 9 Å/s. As the RF etch amount increased from 50–100 Å, the P active resistance increased by 0.5–1 Ω. The resistance also increased as the amount of IMP Ti deposition increased to 300 Å. A measurement of the borderless contact junction leakage current indicated that the current in the P + N well increased by more than an order of magnitude when IMP Ti 250 Å or more was deposited. The contact resistance value was 0.5 Ω. An AC bias improved the IMP Ti deposition rate by 10% in bottom coverage, but there was no significant difference in contact resistance. In the case of applying IMP TiN, the overall contact resistance decreased to 2 Ω compared to CVD TiN, but the distribution characteristics were poor. The best results were obtained under the conditions of RF etch 50 Å, IMP Ti 200 Å, and CVD TiN 2 × 50 Å.

## 1. Introduction

In this report, we sought to find the ideal conditions and device characteristics based on the process splits that were performed for the contact plug process setup of 0.15 μm logic devices. The contact design rule for a 0.15 μm logic device requires a hole size of 0.18 μm and an interlayer dielectric (ILD) thickness of 7500 Å, with an aspect ratio of 4.16:1, making it more fragile than the 3.69:1 (0.23 μm, 8500 Å) of a 0.18 μm logic device. As a result, a borderless contact (BLC) structure was adopted. BLC nitride was used as an etch stop layer and cobalt (Co) silicide [1] was applied to the salicide process [2]. 

As mentioned above, the contact plug design [3] reduction of the 0.15 μm logic device was the main change compared to the 0.18 μm logic device, and this was set up based on the possibility of process change [4]. We examined the physical and electrical characteristics of pre-cleaning, adhesion layer, barrier layer, etc. currently applied to logic devices, and judged whether they were appropriate here, before selecting the optimal contact plug conditions. As with the via plug, the tungsten (W) plug process [5] was applied to the contact plug, and the barrier film and adhesion layer were applied to Metal Organic Chemical Vapor Deposition (MOCVD) TiN [6] and ionized metal plasma (IMP) Ti, respectively [7,8]. Currently, the Ti deposition process that can be applied in terms of bottom coverage utilizes only the IMP method, while in the case of the TiN deposition process, MOCVD [9] TiN, which has excellent bottom coverage characteristics [10], is applied. In addition, Radio Frequency (RF) etch cleaning was adopted for the pre-cleaning process, according to the BLC structure [11,12]. Recently, the trend of standard CMOS processes is to apply Co silicide or Ni silicide in 10nm contact process and copper contact in devices requiring high speed. Future research is also underway to apply nanowires and graphene [13]. However, W is applied to memory devices that do not require high speed, and optimized process conditions are applied according to ILD thickness and contact profile. Therefore, it is important to note that the contact profiles formed through photo lithography and etch processes of all companies are very different in terms of process integration. Constant process development is still in progress. Therefore, in this study, we also established contact plug optimization conditions based on contact resistance and junction leakage characteristics in order to establish optimized contact plug conditions according to RF etching and deposition characteristics according to the given process integration profile.

This report, therefore, is focused on RF pre-cleaning [14], IMP Ti, chemical vapor deposition (CVD) TiN, and CVD W [15] 0.15 μm logic devices based on contact resistance, leakage, etc. after checking basic properties such as the bottom coverage and W filling status of each film. This study attempts to examine and select the optimal conditions for this contact plug process [16].

## 2. Experimental Procedure

The equipment used for depositing the barrier film before the W deposition was the C-03-12 (Endura, AMAT) of the FAB 4 line. This equipment comprises the RF etch (pre-clean), IMP Ti, IMP TiN, CVD TiN chambers, and it is possible to continuously process RF etch–IMP Ti–CVD TiN (or IMP TiN) without an air break. Table 1 shows the specific resistance of each film and Table 2 shows the deposition conditions. The resistivity was obtained by depositing 5000 Å of PETEOS, 200 Å of IMP Ti, 100 Å of CVD TiN, and 300 Å of IMP TiN on the bare wafer. RF etch conditions were performed with RF power at 400 W, RF second power at 275 W, and Ar gas 10 sccm. CVD tungsten deposition was performed using the Concept-1 (Novellus). The air flow of deposition conditions were SiH_4_, 0.025 slm; H_2_, 6 slm; WF 6, 0.28 slm; temp, 395 °C, and SiH_4_ reduction time of the W seed formation step set to 12 s. Under these conditions, the resistivity of tungsten was measured by determining the *R*s using an omnimap after depositing about 3800 Å of tungsten after measuring the resistivity of the barrier film. Step coverage and filling of the CVD TiN, IMP TiN, and CVD W thin films were analyzed using SEM and TEM. Due to the reduction of the contact hole size, it was difficult to obtain the correct cutting surface in general specimen preparation, so the cross section was checked again using the Precision Etching Coating System (PECS) [17]. Existing published paper [18] support this paper.

## 3. Results and Discussion

As shown in Table 1, the uniformity of the IMP process tended to be high, a tendency known as a characteristic of the IMP process rather than a problem of the deposition conditions. CVD TiN was deposited at 2 × 50 Å for a total of 100 Å and IMP TiN was deposited without an AC bias (as with IMP Ti). The CVD TiN 2 × 50 process is a two-step process of film deposition and in situ plasma post-treatment. The reason why the IMP TiN process was compared with CVD TiN was to examine the possibility of optimizing an integrated process because of IMP TiN’s advantages and its low resistivity in terms of throughput [19].

First, each film was deposited using wafer defining contact holes to confirm the bottom coverage of IMP Ti and CVD TiN. The contact holes were defined in accordance with the 150 nm-class logic contact design rules [20,21]. Figure 1 shows the bottom coverage of IMP Ti and CVD TiN. The bottom CD of the contact hole implemented on the test wafer with TEM was about 0.185 μm and the ILD thickness was about 7200 Å, while the aspect ratio was calculated to be about 3.89:1. As a result of measuring the deposition amount at the hole bottom of each film, it was confirmed that about 90 Å was deposited on the contact hole bottom when IMP Ti 220 Å was deposited, and about 80 Å was deposited when CVD TiN 105 was deposited. In IMP Ti was about 40.9%, CVD TiN was about 76.2%. From these results, the bottom coverage in terms of the aspect ratio in the design contact hole is considered to be good, and there was no problem in applying it to the 150 nm logic device process integration process [22,23]. The electrical properties such as contact resistance were measured, in order to verify this.

To confirm the W filling state in the contact hole, CVD tungsten deposition proceeded for 12 s of SiH4 [24] reduction time, a W seed formation step [25,26]. Under the same conditions as in the measurement of the resistivity of the barrier film, the resistivity calculated after deposition of tungsten 3800 Å was about 11.5 Ω·cm. Figure 2 shows an SEM image of the contact plug cross section. As shown in the figure, the tungsten filling of the contact hole was confirmed, and it is thought that the tungsten could be filled to ILD 6700 Å, even with a contact hole size of 0.14 μm, under the current deposition conditions. Considering the 0.14 μm hole size, it was confirmed that there was no problem in filling the contact hole of 150 nm-class logic devices. Interlayer dielectric (ILD) layers of 150 nm-class logic devices are stacked from the bottom in the order of nitride–BPSG (borophosphosilicate glass) –PETEOS (plasma enhanced tetraethylorthosilicate layers) [26,27]. Due to this structure, there is a problem in applying pre-cleaning to wet clean the substrate before barrier deposition [28]. That is, there are problems of deformation of the hole profile due to the selective etching ratio of each film and problems of characteristic leakage deterioration in the BLC structure [29]. As a result, a pre-cleaning method using an RF etch is used. However, RF etch cleaning has a high salicide etching rate [30], and the salicide etching amount at the hole bottom was confirmed by the RF etch quantity. The test wafer had a contact hole height of about 6700 Å and a bottom CD of about 0.19 μm. The aspect ratio was about 3.53. The deposition thickness of BLC nitride was 200 Å and the salicide thickness was about 350~400 Å. The RF etch conditions were RF power at 400 W, RF second power at 275 W, Ar gas 10 sccm, and were split into 50 Å, 100 Å, and 150 Å layers, respectively, based on the thermal oxide on the plate. After coating, cross-sectional SEM observations made it easy to check the salicide thickness. The results are shown in Figure 3.

As shown in Figure 3, the amount of salicide remaining at the bottom of each hole was measured at about 250 Å, 200 Å, and 130 Å as the etch amount increased, and the amount of salicide was measured at about 100 Å, 150 Å, and 220 Å, respectively. It is difficult to accurately compare the thicknesses because salicide formation varies locally, but the salicide etch rate at an aspect ratio of 3.53 was calculated to be about 13–18 Å/s, versus the RF etch rate of 9 Å/s on the plate.

In order to evaluate the contact resistance characteristics according to the process conditions, IMP Ti and CVD TiN conditions were fixed, RF etch was fixed at 50 Å, 70 Å, 100 Å, RF etch and CVD TiN conditions were fixed, and IMP Ti thickness was 150 Å, 200 Å, 250 Å, 300, Å respectively.

Figure 4 shows the results of contact resistance by experimental conditions to understand the contact resistance and leakage characteristics according to the barrier deposition scheme. As can be seen, N active resistance shows poor results, while P active resistance shows 8–12 Ω for chain resistance, and 6–8 Ω for kelvin resistance, without any difference between the splits. The reason for this N active resistance was found to be due to the abnormal oxidation phenomenon on the N active side of the salicide. Specimens were treated with a buffered oxide etch (BOE) [31] process for 15 s after cross section cutting and observed by cross section using SEM (Figure 5). A trace of oxide-like material was etched off just below the contact hole, which may have led to an increase in resistance. In the P active area, this phenomenon did not occur [32], so the contact resistance comparison was made only through the P active area [33]. The resistance and leakage values in the wafer that was processed after the salicide process had stabilized, were compared again. As for the P etch resistance, we found that as the amount of RF etch increased from 50 to 100 Å, the resistance also increased by about 0.5–1.0 Ω, though the distribution tended to get worse [34]. However, in the case of IMP Ti 150 Å, while the distribution of kelvin resistance deteriorated, the chain pattern was similar to those under other conditions.

Figure 6 shows the P active BLC contact resistance values by split. Overall, there was no condition showing a bad resistance value. However, as with the kelvin resistance and the chain resistance, the resistance increase was 1–1.5 Ω as the RF etch quantity and IMP Ti deposition thickness were increased.

Figure 7 shows contact P + N BLC leakage according to the RF etching amount and IMP Ti deposition thickness. In general, the leakage characteristics were poor, especially when the IMP Ti deposition thickness was more than 250 Å, the difference was more by as much as an order of magnitude. There were no significant differences in the other splits, and there was no difference between the end type and the peri type. Based on the above results, P active contact leakage characteristics of IMP Ti thickness of 250 Å or more are difficult to assume in terms of leakage, though the lower the RF etch and IMP Ti thickness, the better the resistance characteristics [35].

Based on the above results, the RF etch was conducted at 50 Å, 30 Å, and IMP Ti at 200 Å and 150 Å, respectively, and IMP TiN was applied to compare to the process proficiency of CVD TiN. Furthermore, to evaluate the damage caused by the AC bias on the IMP Ti deposition, the conditions of deposition by applying an AC bias of 200 W were also performed. The contact kelvin resistance is shown in Figure 8 and the chain resistance in Figure 9. The kelvin resistance was 6–8 Ω on the active side, and 5–7 Ω on the poly side, and the chain resistance ranged from 9–11 Ω on the active side, and 8–10 Ω on the poly side. Compared to the poly side, the overall contact resistance was about 0.5 Ω [36]. According to the experimental conditions, with the RF etch and IMP Ti reduced to 30 Å and 150 Å respectively, this tended to decrease the contact resistance by about 0.5 Ω, but with RF etch 50 Å and IMP Ti 200 Å, the results showed a stable distribution without deterioration of the contact resistance value.

In the case of IMP Ti deposition, the split with AC bias fixed at 200 Å of IMP Ti deposition thickness, was about 51.2%. In terms of bottom coverage, the effect was about 10% better than that of 40.9% in the ‘no bias’ condition. However, as shown in Figure 8 and Figure 9, in terms of contact resistance, the change of resistance tends to increase slightly or not at all, when an AC bias is used [37].

In the case of depositing IMP TiN 200 µs instead of CVD TiN 100 µs, the kelvin resistance [38] was about 1.5–2 Ω. However, unlike CVD TiN, which showed an even distribution of resistance, the resistance distribution of 10–20% tends to reduce. This phenomenon is different from that shown in the via resistance [39], and it seems that there is a problem in applying IMP TiN to the contact barrier [40]. In the case of CVD TiN, as shown in the via, it was confirmed that the stable distribution was shown in the 2 × 50 condition without much problem. In terms of leakage current distribution, the RF etch 50 Å–IMP Ti 200 Å–CVD TiN 2 × 50 Å seems to be the best condition. It seems that there is little effect and the BLC leakage [29] current is expected to increase when IMP Ti thickness is over 250 Å. The increase in BLC leakage current with increasing IMP Ti thickness is expected due to the deposited Ti reacting with active by diffusing salicide through the rapid thermal process (RTP).

## 4. Summary

Based on the physical and electrical characteristics measured, the RF etch 50 Å, IMP Ti 200 Å, CVD TiN 2 × 50 Å, and CVD W 3800 Å conditions are not considered to be optimal for the 100 nm contact plug. With the RF etch and IMP Ti thickness lowered to 30 Å and 150 Å, respectively, the contact resistance could be further reduced. However, the difference was 0.5 Ω. There was no significant effect in terms of resistance distribution or BLC leakage. On the other hand, the RF etch is expected to reduce the amount to 30, resulting in sufficient contact hole floor cleaning effect and difficulty in securing process margin. Although IMP TiN showed an improved effect in terms of contact resistance, it is difficult to apply due to the poor resistance distribution, though it may be applicable if resistance distribution improvements are made in the future. At an aspect ratio of 3.89:1, IMP Ti bottom coverage of 40.9%, and CVD TiN bottom coverage of 76.2% was confirmed, W filling was confirmed. The salicide etching rate by RF etch was about 13–18 Å at 3.53:1 aspect ratio. At this stage, the etching rate of the thermal oxide plate was 9 Å/s. As the RF etch amount increased from 50 to 100 Å, the contact resistance of P active also increased about 0.5–1 Ω, and the resistance increased slightly too as the IMP Ti deposition increased to 300 Å. As a result of the BLC junction leakage current measurement, a current increase of about an order of magnitude was observed in the P + N well when IMP Ti 250 Å or more was deposited. In terms of the RF etch quantity and IMP Ti deposition thickness, the contact resistance value was reduced by about 0.5 Ω at 30 Å with the RF etch 50 Å and 150 Å than with IMP Ti 200 Å. In the case of IMP Ti deposition, the effect of an AC bias showed a 10% improvement in bottom coverage. There was no significant difference in contact resistance, but rather a slight increase. The overall contact resistance of IMP TiN decreased by 2 Ω compared to CVD TiN, but the distribution characteristics were poor. The best results were obtained under the conditions of RF etch 50 Å, IMP Ti 200 Å, and CVD TiN 2 × 50Å.

## Figures and Tables

**Figure 1 micromachines-11-00170-f001:**
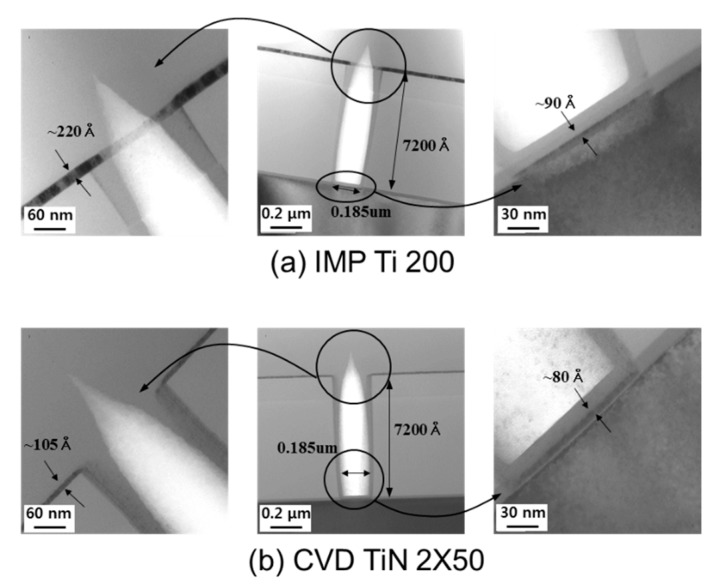
TEM image of ionized metal plasma (IMP) Ti and chemical vapor deposition (CVD) TiN at an aspect ratio of 3.89: 1. (**a**) IMP Ti 200. (**b**) CVD TiN 2 × 50 (unit: Å).

**Figure 2 micromachines-11-00170-f002:**
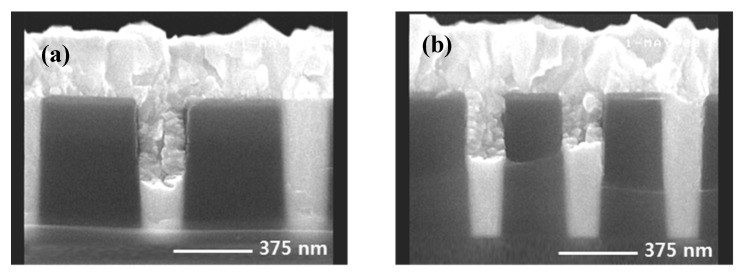
SEM image of cross section after tungsten deposition of contact hole size (**a**) 0.16 µm with interlayer dielectric (ILD) thickness 6700 Å and (**b**) 0.14 µm with ILD thickness 6700 Å.

**Figure 3 micromachines-11-00170-f003:**
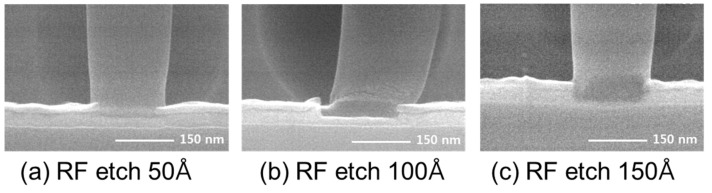
Cross-sectional SEM of etching amount of Co salicide according to Radio Frequency (RF) etch amount. (**a**) RF etch 50 Å. (**b**) RF etch 100 Å. (**c**) RF etch 150 Å.

**Figure 4 micromachines-11-00170-f004:**
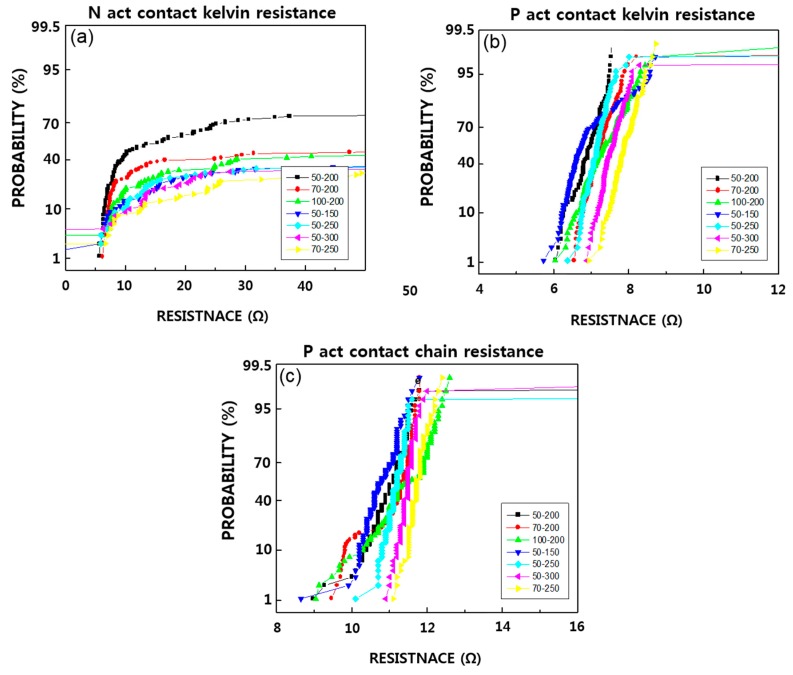
Result of (**a**) N act kelvin, (**b**) P act kelvin, (**c**) P act chain contact resistance by process conditions (unit: Å).

**Figure 5 micromachines-11-00170-f005:**
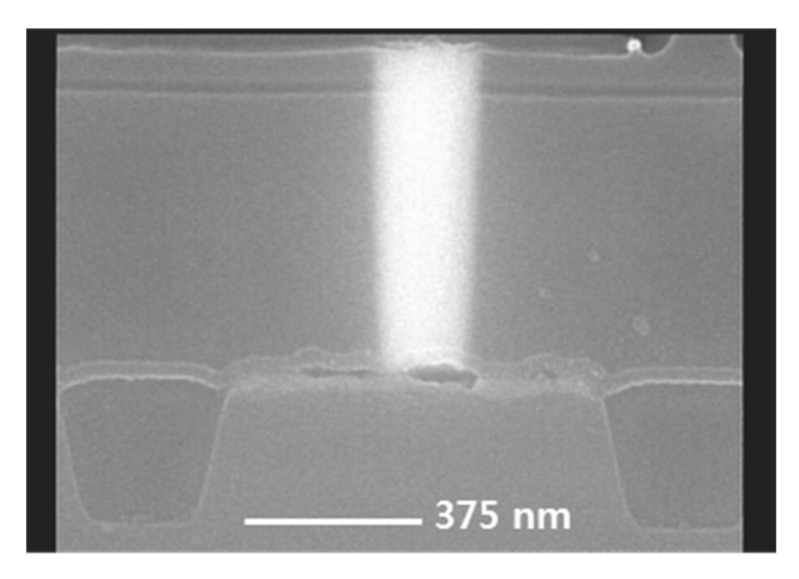
Cross section SEM image of specimens treated with a buffered oxide etch (BOE) process.

**Figure 6 micromachines-11-00170-f006:**
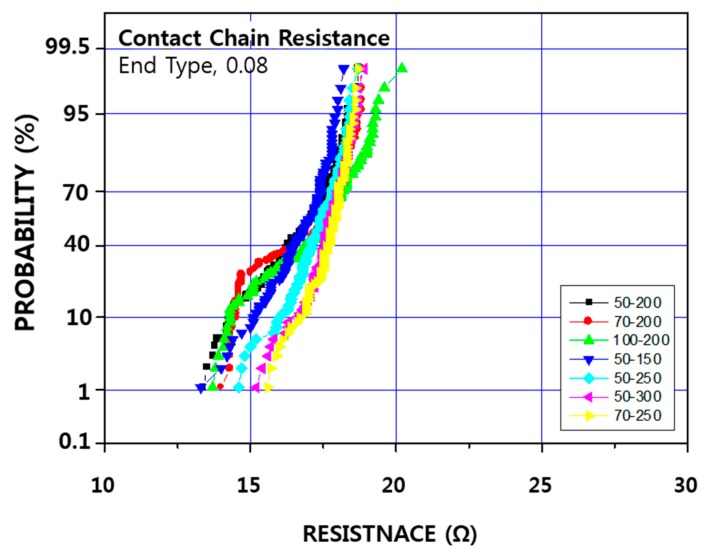
Active BLC contact resistance according to RF etch amount and IMP Ti deposition thickness. (unit: Å).

**Figure 7 micromachines-11-00170-f007:**
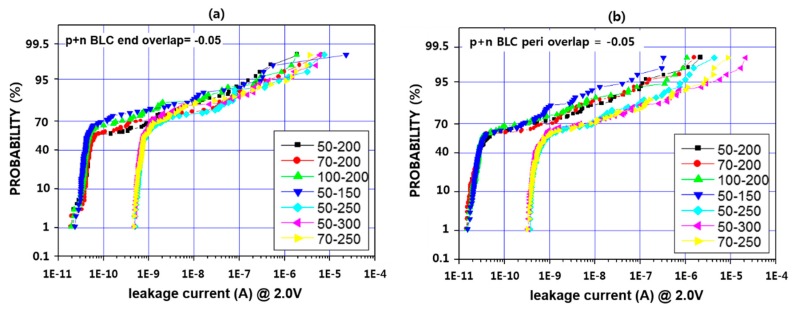
Contact P + N BLC leakage at overlap −0.05 according to RF etch amount and IMP Ti deposition thickness. (**a**) end type (**b**) peri type (unit: Å).

**Figure 8 micromachines-11-00170-f008:**
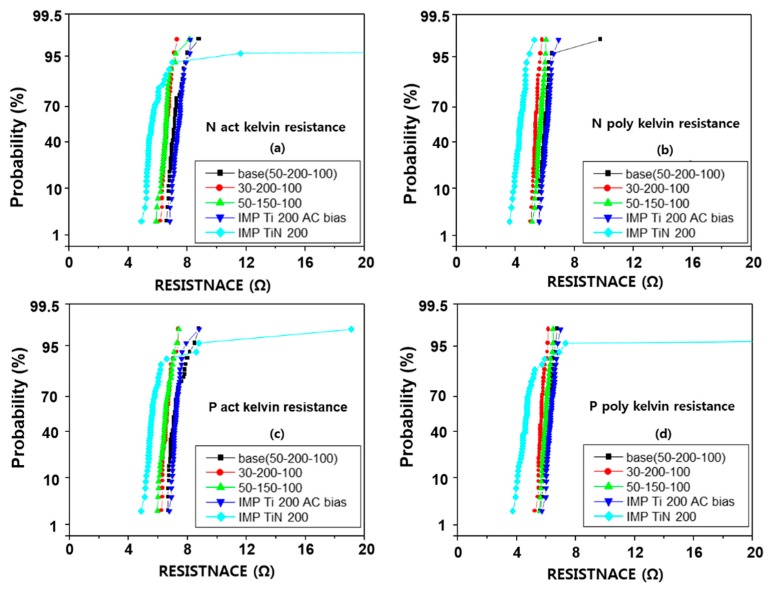
Electrical properties of (**a**) N act kelvin, (**b**) N poly kelvin, (**c**) P act kelvin, and (**d**) P poly contact resistance (Kelvin type) (unit: Å).

**Figure 9 micromachines-11-00170-f009:**
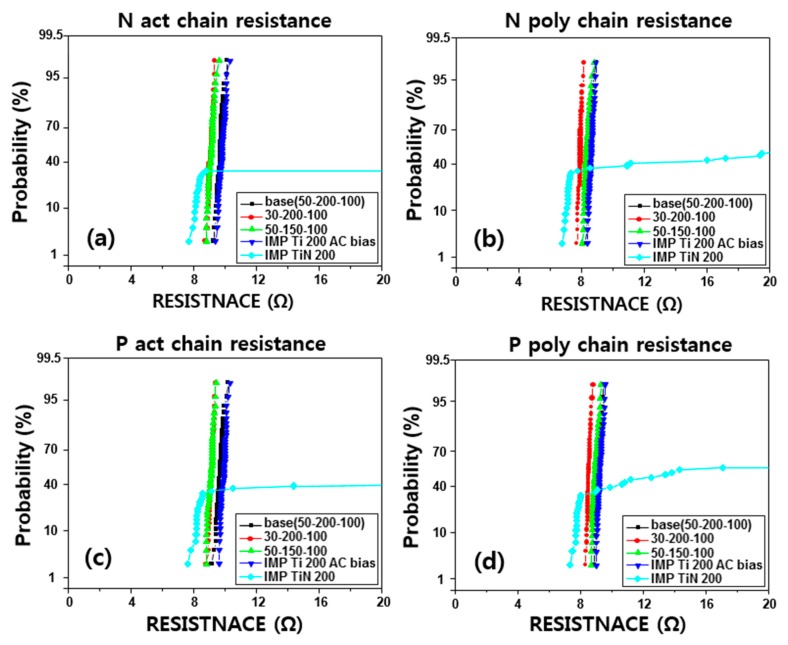
Electrical properties of (**a**) N act chain, (**b**) N poly chain, (**c**) P act chain, and (**d**) P poly chain contact resistance (chain type) (unit: Å).

**Table 1 micromachines-11-00170-t001:** Sheet resistance (*R*s) and specific resistance value (*ρ*) for each film.

	IMP Ti	CVD TiN	IMP TiN
Average Rs (Ω/cm^2^)	39.9	302.05	29.06
Unif. (%)	4.99	3.55	9.05
TEM thickness (center, Å)	195	100	345
*R*s (center, Ω/cm^2^)	37.5	280.8	24.73
Specific Resistance (μΩ·cm)	~73	~280	~85
Stress (dyne/cm^2^)	−2.142 × 10^9^	−4.788 × 10^9^	−6.762 × 10^9^

**Table 2 micromachines-11-00170-t002:** Deposition conditions for each film.

**IMP Ti**	**DC power**	**RF power**	**AC bias**	**Ar**		
	2250 W	2750 W	0 W	56 sccm		
**CVD TiN**	**DEP**				**Plasma Treat**	
	Pressure	TEMP	He carr	Pressure	RF power	TEMP
	1.5 Torr	450 °C	225 sccm	1.3 Torr	750 W	450 °C
**IMP TiN**	**DC power**	**RF power**	**AC bias**	**Ar**	**N2**	
	4000 W	2500 W	0 W	25 sccm	28 sccm	

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
