# Peer review of "Effect of Contact Plug Deposition Conditions on Junction Leakage and Contact Resistance in Multilevel CMOS Logic Interconnection Device"

_micromachines, 2020, doi:10.3390/mi11020170_

Round 1

Reviewer 1 Report

The paper studies effect of different deposition conditions on junction leakage and contact resistance in interconnection for CMOS logic device. The authors demonstrate significant amount of work on deposition and characterization of via plug. However, the paper has several shortcomings need to be corrected. The writing of this paper needs more effort and some figures (like Figure 4) are unreadable. Some foreign character appearing in the context (line 66) is not appropriate. Furthermore, I found some references show no obvious relation to the context where they are cited, like reference 15 and 16 in line 54. Since the paper mainly discusses the effect of different deposition conditions on junction leakage and contact resistance rather than the effect of junction leakage and contact resistance, the title should be changed. Thus, I think the manuscript is not suitable for publication in Micromachines before a thorough modification is made.

Author Response

Response to Reviewer 1’s Comments

Point 1: The paper studies effect of different deposition conditions on junction leakage and contact resistance in interconnection for CMOS logic device. The authors demonstrate significant amount of work on deposition and characterization of via plug. However, the paper has several shortcomings need to be corrected. The writing of this paper needs more effort and some figures (like Figure 4) are unreadable. Some foreign character appearing in the context (line 66) is not appropriate. Furthermore, I found some references show no obvious relation to the context where they are cited, like reference 15 and 16 in line 54. Since the paper mainly discusses the effect of different deposition conditions on junction leakage and contact resistance rather than the effect of junction leakage and contact resistance, the title should be changed. Thus, I think the manuscript is not suitable for publication in Micromachines before a thorough modification is made. 

Response 1: We thank you for considering our submission and for your valuable comments. The manuscript has been carefully revised according to the reviewer’s comments. We have addressed the reviewer’s comments individually below.

The reviewer pointed out that the revision was completed. We have also revised the reference pointed out by the reviewer. The review title pointed out by the reviewer has been revised as follows.

Effect of Junction Leakage and Contact Resistance on the Contact Plug Process Parameters in Multilevel CMOS Logic Interconnection Device

the effect of contact plug deposition conditions on junction leakage and contact resistance in Multilevel CMOS Logic Interconnection Devi

Reviewer 2 Report

This paper mainly discusses the effect of junction leakage and contact resistance of CMOS in the contact plug process.
Despite many experiments performed in this article, the research is not significantly innovative and compared with today's technology.
The contact plug process already has quite mature products in the CMOS process. The dissertation should be compared with the contact resistance and junction leakage of the current major CMOS processes (such as the processes from TSMC, UMC, Samsung, etc.). The research should address existing CMOS technologies, explain the differences between them, and prove its excellence. It is recommended to modify this paper based on this point.

The Figures in Fig.4, Fig.7 and Fig.9 are too vague. Please clearly indicate the numerical units in all figures.

Author Response

Response to Reviewer 2’s Comments

Point 1: This paper mainly discusses the effect of junction leakage and contact resistance of CMOS in the contact plug process.
Despite many experiments performed in this article, the research is not significantly innovative and compared with today's technology.
The contact plug process already has quite mature products in the CMOS process. The dissertation should be compared with the contact resistance and junction leakage of the current major CMOS processes (such as the processes from TSMC, UMC, Samsung, etc.). The research should address existing CMOS technologies, explain the differences between them, and prove its excellence. It is recommended to modify this paper based on this point.

Response 1: We thank you for considering our submission and for your valuable comments. The manuscript has been carefully revised according to the reviewer’s comments. We have addressed the reviewer’s comments individually below.

As the reviewer pointed out, the contact plug process in the current CMOS process is quite mature. It is also a process adopted by most major companies such as Samsung, SK Hynix, TSMC, and UMC. However, it is important to note that the contact profiles formed through photo lithography and etch processes of all companies are very different in terms of process integration. Constant process development is still in progress. Therefore, in this study, we also established contact plug optimization conditions based on contact resistance and junction leakage characteristics in order to establish optimized contact plug conditions according to RF etching and deposition characteristics according to the given process integration profile. I think there is.

Point 2: The Figures in Fig.4, Fig.7 and Fig.9 are too vague. Please clearly indicate the numerical units in all figures.

Response 2: As reviewer pointed out, the revision was completed.

Reviewer 3 Report

The paper "Effect of Junction Leakage and Contact Resistance on the Contact Plug Process Parameters in Multilevel CMOS Logic Interconnection Device" presents the ideal conditions and device characteristics based on the process splits that were performed for the contact plug process setup of 0.15 μm logic devices. The authors examined the physical and electrical characteristics of pre-cleaning, adhesion layer, barrier layer and currently applied to logic devices.

The work is well structured and presents novelty elements but to be published I think, the authors need to make some corrections and improvements.

It was introduce directly in Abstract terms such as W, MLM, IMP TiN, RF, CVD, without giving any explanation. Normally these terms need to be defined first, before they can be abbreviated. In the abstract the authors introduce the purpose of the work too directly without explaining the necessity of this research.

In the Experimental part, the thin films were analyzed using SEM and TEM in cleen conditions or in air? The electrical properties such as contact resistance were measured also in in the room where they were deposided? It is not clear from the forms. Can the studied phenomenon be reproduced with the help of simulations that confirm the choice of suitable manufacturing parameters?

How the "probability" was calculated from figure 4? It is not understood from the context how this representation was reached. Probability cannot be measured experimentally. I believe that arbitrary values ​​should be set here, clarifications are needed in the text.

In Conclusion part the authors observe that the overall contact resistance of IMP TiN decreased by 2 Ω compared to CVD TiN, but the distribution characteristics were poor. The best results were obtained under the conditions of RF etch 50 Å, IMP Ti 200 Å and CVD TiN 2X50Å. The authors can justify in practice the results? Are the experiments performed reproducible? What happens if the parameters used are completely different?

Best regards,

Author Response

Response to Reviewer 3’s Comments

Point 1: The paper "Effect of Junction Leakage and Contact Resistance on the Contact Plug Process Parameters in Multilevel CMOS Logic Interconnection Device" presents the ideal conditions and device characteristics based on the process splits that were performed for the contact plug process setup of 0.15 μm logic devices. The authors examined the physical and electrical characteristics of pre-cleaning, adhesion layer, barrier layer and currently applied to logic devices.

The work is well structured and presents novelty elements but to be published I think, the authors need to make some corrections and improvements.

It was introduce directly in Abstract terms such as W, MLM, IMP TiN, RF, CVD, without giving any explanation. Normally these terms need to be defined first, before they can be abbreviated. In the abstract the authors introduce the purpose of the work too directly without explaining the necessity of this research.

Response 1: We thank you for considering our submission and for your valuable comments. The manuscript has been carefully revised according to the reviewer’s comments. We have addressed the reviewer’s comments individually below.

We revised terms such as W, MLM, IMP TiN, RF, CVD in manuscript as follows

W(Tungsten), MLM(Multi-Level Metallization), IMP TiN (Ionized Metal Plasma Deposition) TiN, RF(Radio Frequency),  CVD(Chemical Vapor Deposition)

Point 2:  In the Experimental part, the thin films were analyzed using SEM and TEM in cleen conditions or in air? The electrical properties such as contact resistance were measured also in in the room where they were deposided? It is not clear from the forms. Can the studied phenomenon be reproduced with the help of simulations that confirm the choice of suitable manufacturing parameters? How the "probability" was calculated from figure 4? It is not understood from the context how this representation was reached. Probability cannot be measured experimentally. I believe that arbitrary values ​​should be set here, clarifications are needed in the text.

Response 2: Thank you for your comment.

SEM and TEM analyzes are made of specimens in an air atmosphere and the analysis is a vacuum chamber. Electrical characterization was performed continuously in-line after all deposition processes were completed. No separate simulation was performed and the probability of PCM data means the distribution of resistance change in the total number of dies measured. In other words, in the semiconductor process, PCM data is usually expressed like this.

Point 3:  In Conclusion part the authors observe that the overall contact resistance of IMP TiN decreased by 2 Ω compared to CVD TiN, but the distribution characteristics were poor. The best results were obtained under the conditions of RF etch 50 Å, IMP Ti 200 Å and CVD TiN 2X50Å. The authors can justify in practice the results? Are the experiments performed reproducible? What happens if the parameters used are completely different?

Response 3: We thank you for considering our submission and for your valuable comments. We believe that the optimized process through the present experiments has a very high reproducibility and is currently applied to the mass production process as a basis for supporting it.

Round 2

Reviewer 1 Report

Figure 2 presents two SEM images but without appropriate labelling and figure caption. In Fig. 4-9, authors used legend like 50-200 or 30-200-100 but without correlating these numbers to the experiment conditions. Authors claimed that they correct the reference issue. But, updated reference 15 and 16 has no relation to the sentence where they are cited. The foreign character in line 69 still exists. Most importantly, authors used exact same experimental details as they published in a recent paper (Cui Y. et al, Process Optimization of Via Plug Multilevel Interconnections in CMOS Logic Devices) without reference. Thus, the manuscript is not recommended for acceptance before correcting all issues mentioned above.

Reviewer 2 Report

I still recommend comparing innovation to the standard CMOS processes in today's mainstream fabs such as TSMC, samsumg, globalfoundry. At least, the improvement for today's technology should be addressed and compared in the introduction.

The units in Figures 4, 7, 8 and 9 are still not clearly marked

Round 3

Reviewer 1 Report

The manuscript is suitable for publication in Micromachines.

Reviewer 2 Report

  The paper is acceptable, I have no other questions.